# Current Trends of Essential Trace Elements in Patients with Chronic Liver Diseases

**DOI:** 10.3390/nu12072084

**Published:** 2020-07-14

**Authors:** Takashi Himoto, Tsutomu Masaki

**Affiliations:** 1Department of Medical Technology, Kagawa Prefectural University of Health Sciences, 281-1, Hara, Mure-Cho, Takamatsu, Kagawa 761-0123, Japan; 2Department of Gastroenterology and Neurology, Kagawa University School of Medicine, Kagawa 761-0123, Japan; tmasaki@med.kagawa-u.ac.jp

**Keywords:** zinc, selenium, copper, iron, chronic hepatitis, liver cirrhosis, nonalcoholic fatty liver disease, autoimmune liver disease, hepatic fibrosis, hepatic steatosis

## Abstract

Essential trace elements play crucial roles in the maintenance of health, since they are involved in many metabolic pathways. A deficiency or an excess of some trace elements, including zinc, selenium, iron, and copper, frequently causes these metabolic disorders such as impaired glucose tolerance and dyslipidemia. The liver largely regulates most of the metabolism of trace elements, and accordingly, an impairment of liver functions can result in numerous metabolic disorders. The administration or depletion of these trace elements can improve such metabolic disorders and liver dysfunction. Recent advances in molecular biological techniques have helped to elucidate the putative mechanisms by which liver disorders evoke metabolic abnormalities that are due to deficiencies or excesses of these trace elements. A genome-wide association study revealed that a genetic polymorphism affected the metabolism of a specific trace element. Gut dysbiosis was also responsible for impairment of the metabolism of a trace element. This review focuses on the current trends of four trace elements in chronic liver diseases, including chronic hepatitis, liver cirrhosis, nonalcoholic fatty liver disease, and autoimmune liver diseases. The novel mechanisms by which the trace elements participated in the pathogenesis of the chronic liver diseases are also mentioned.

## 1. Introduction

The liver plays indispensable roles in the maintenance of essential trace elements homeostasis [1,2]. Most of the trace elements are absorbed from the duodenum and/or jejunum and flow out in the portal circulation by binding to the plasma proteins. These trace elements are distributed to the tissues or organs that require them. It is primarily the liver that initiates the synthesis of the proteins bound for several trace elements, including zinc (Zn), selenium (Se), iron (Fe), and copper (Cu) in order to transport or distribute these trace elements. The liver is also involved in the excretion of trace elements such as Cu and magnesium (Mg), since the liver acts as a producer of bile. 

Most of the trace elements that have immunomodulatory and antimicrobial activities generally serve as enzyme-cofactors, antioxidants, and/or anti-inflammatory agents [3]. Impairments of the liver function result in disturbances of the metabolism of trace elements, leading to the initiation of oxidative stress and the subsequent inflammatory and/or fibrotic changes in the liver. The impairment of the homeostasis of trace elements leads to various inflammatory changes and/or metabolic abnormalities such as those observed in inflammatory bowel disease [4], diabetes mellitus [5], dyslipidemia [6], and sarcopenia [7] as well as chronic liver injuries.

Recent advances in molecular biological techniques have enabled us to elucidate the novel mechanisms by which the impairment of trace elements metabolism causes these metabolic abnormalities. Genetic polymorphisms to regulate the metabolism of some trace elements have been identified. The alteration of gut flora results in disorders of some trace elements metabolism, exacerbating hepatic steatosis and/or fibrosis. Some microRNAs may participate in the metabolism of a trace element. 

This review highlights the current trends of four essential trace elements (Zn, Se, Fe, and Cu) in chronic liver diseases (CLDs), including chronic hepatitis, liver cirrhosis, nonalcoholic fatty liver disease (NAFLD), and autoimmune liver disease. We also discuss the novel mechanisms by which impairment of the metabolism of trace elements may account for the pathogenesis of these CLDs. 

## 2. The Roles of Trace Elements 

### 2.1. Translation, Transcription, and Replication of Hepatitis Viruses 

Some trace elements are required to evoke an effective immune response to viral infections, whereas other trace elements are involved in the clearance of viruses. The inhibitory roles of these essential trace elements are revealed in the translation, transcription, and replication of hepatitis viruses, including hepatitis B virus (HBV), hepatitis C virus (HCV), and hepatitis E virus (HEV). Se and Zn were demonstrated to suppress the transcription and replication of hepatitis viruses [8,9]. Table 1 shows the function of such trace elements in the translation, transcription, and replication of these viruses.

#### 2.1.1. HBV 

HBV is a member of the *Hepadnaviridae* family and has circular and partially double-stranded DNA. A persistent HBV infection causes major public health problems throughout the world, especially in East Asia and Africa [23]. Individuals who are infected with HBV can develop chronic liver disease, including chronic hepatitis and liver cirrhosis, and subsequently hepatocellular carcinoma (HCC). 

The Zn status determined the responsiveness to HBV vaccination. The responsiveness was evaluated by the serum anti-HBs level. The serum anti-HBs level was markedly decreased in rats fed a diet with lower Zn content [10]. The poor response to HBV vaccination in such rodents may be derived from suppressed T-lymphocyte proliferation due to Zn deficiency. Zn deficiency affects both the innate and adaptive immune systems, leading to the impaired activation and maturation of lymphocytes [24]. Likewise, gestational Zn deficiency in a murine model resulted in weaker responsiveness to HBV vaccination in offspring mice because of the decreased number of B cells and impaired HBV-specific IgG production [11]. 

In contrast, Cheng et al. demonstrated that sodium selenite suppressed HBV protein expression, transcription, and genome replication, using the human hepatoma cell lines [25]. Se is likely to activate p53 by promoting its expression and phosphorylating multiple sites, and suppressing the activities of HBV promoters and enhancers. 

#### 2.1.2. HCV 

HCV is a positive-polarity, single-stranded RNA virus that belongs to the *Flaviviridae* family. HCV infects hepatocytes and is usually transmitted through exposure to infected body fluids, including blood transfusions and drug abuse. Chronic HCV infection is thought to cause the production of reactive oxygen species (ROS) and subsequently inflammation and fibrosis in the liver, leading to chronic hepatitis, liver cirrhosis, and ultimately to HCC [26]. A persistent HCV infection also results in various metabolic abnormalities such as insulin resistance, hepatic steatosis, dyslipidemia, and Fe overload [27].

A large amount of evidence obtained over the past decades indicates that Zn plays suppressive roles in the replication of HCV. Yuasa et al. demonstrated that Zn salts acted as a negative regulator of the virus replication in genome-length HCV RNA-replicating cells [12]. The authors speculated that Zn might affect the NS2 or NS3 protein and consequently inhibit the replication of genome-length HCV RNA. In later studies, the N2/3 auto-cleavage activity and NS3 protease activity were both confirmed to be Zn-dependent [28], which seems to be plausible because NS3 protease is one of the Zn-containing enzymes [29]. 

It is of interest that zinc sulfate can reduce the HCV replication in vitro [13]. However, this effect of zinc sulfate was reduced when metallothionein (MT) genes were knocked out [13], suggesting that MTs are either directly antiviral by sequestering Zn away from viral MTs such as NS5 [30], or indirectly antiviral by acting as Zn chaperones and facilitating antiviral signaling [31]. It is reasonable that serum Zn levels in patients with HCV-related CLD were increased by treatment with direct-acting antiviral agents (DAAs) [32]. Read et al. recently elucidated that a single-nucleotide polymorphism of interferon-lambda 3 (IFN-λ3), which is one of the antiviral and pro-inflammatory cytokines, was correlated with increased hepatic MT expression through increased systemic Zn levels [14]. 

A decline in the systemic Se concentration may be attributable to an intracellular replication of HCV. RNA viruses, including HCV and human immunodeficiency virus (HIV), encode a Se-dependent glutathione peroxidase (GPx) module [15], which is one of the selenoproteins and protects against damage induced by free radicals. It is plausible that serum Se levels were negatively correlated with the loads of HCV RNA in patients with chronic hepatitis C [33], but our previous study did not confirm this phenomenon in patients with HCV-related CLD [34]. The decrease in a circulating Se concentration may also reflect a systemic inflammatory response [35]. Notably, Murai et al. demonstrated that the hepatic selenoprotein P mRNA was upregulated by an HCV infection, and that its knockout in hepatocytes caused an induction of IFN-stimulated genes and a subsequently inhibited the replication of HCV [16]. 

It remains controversial whether Fe enhances the replication of HCV [17,18]. However, Fe proved to facilitate the translation of HCV by stimulating the expression of eukaryotic initiation factor 3A (elF3A) [19]. The expression of hepcidin, a negative regulator of Fe, was suppressed in HCV-infected cells [36]. This suppressive effect may be regulated by histone acetylation. Foka et al. also documented that the overexpression of hepcidin antimicrobial peptide (HAMP) promoted the translation of HCV in vitro [20]. 

No evidence has been accumulated showing that Cu plays a crucial role in the translation or replication of HCV. However, cuprous oxide, a variant form of Cu, is likely to inhibit the entry of HCV pseudoparticles into hepatic cells without any effect on the replication of HCV [21]. Metallopeptide Cu-GGHYrFK, which targets stem loop IV (SLIV) of the HCV internal ribosome entry site (IRES) is expected to become a distinct therapeutic agent for the treatment of an HCV infection [37].

#### 2.1.3. HEV

HEV is a single-stranded RNA virus that belongs to the *Hepeviridae* family. Acute hepatitis caused by an HEV infection is usually self-limiting in healthy subjects. However, a persistent HEV infection can progress to chronic hepatitis or even to liver cirrhosis in immunocompromised individuals [38].

It was recently elucidated that Zn salts (zinc sulfate and zinc acetate) directly inhibited the activity of viral RNA-dependent RNA polymerase, leading to the inhibition of an HEV replication [22]. In addition, a nonstructural HEV protein was currently identified as a putative Zn-binding protein [39].

### 2.2. The Roles of Trace Elements in Hepatic Inflammation

#### 2.2.1. Zn 

Zn is widely known to have cytoprotective properties against oxidative stress, apoptosis, and inflammation [40,41,42]. Zn deficiency thus causes the production of ROS, and subsequently leads to inflammation in the liver (Figure 1). Zn deficiency is often observed in patients with chronic hepatitis [33,43,44,45,46,47,48,49,50] or in those with NAFLD [51] (Table 2), although a lower Zn concentration does not necessarily correspond to the inflammation process [2]. The serum Zn levels of patients with chronic hepatitis were inversely correlated with serum transaminase levels [47] and with their histological activity scores [45]. 

Zn supplementation has a favorable effect on hepatic inflammation in such patients. Our previous study revealed that the additional administration of polaprezinc (225 mg/day for six months), a complex of Zn and L-carnosine, improved serum transaminase levels in patients with HCV-related CLD, including chronic hepatitis and liver cirrhosis, by attenuating hepatic Fe storage [67] (Table 3). Matsumura et al. also confirmed the effect of polaprezinc (150 mg/day for three years) on hepatic inflammation in patients with chronic hepatitis C [68]. Zn is also administered as an antioxidant adjuvant to IFN in patients with chronic hepatitis C. The combination treatment of IFN with antioxidants, including Zn [69,70,71] and N-acetylcysteine, sodium selenite, and vitamin E [72] had promising results in such patients, although a meta-analysis did not demonstrate the beneficial effects by administration of Zn [73].

#### 2.2.2. Se 

Se is a constituent of GPx which protects against the damage induced by ROS [9,83]. Reduced serum Se levels are frequently observed in patients with chronic hepatitis [34,44,48,50,54,55,84,85,86]. Serum GPx levels were also lower in patients with chronic hepatitis C than those in cases of a normal healthy control [17,78,87] (Table 2). However, we did not find an inverse correlation between the serum Se and alanine aminotransferase (ALT) level in patients with HCV-related CLD [34]. It is of interest that a Zn transporter, ZIP 8 was associated with Se homeostasis, and that a decrease in the ZIP 8 activity due to Se deficiency potentially evoked liver injury [86]. 

The effects of Se administration on hepatic inflammation were examined in diabetic rats fed a Zn-deficient diet. The Se supplementation (sodium selenite 0.5 mg/kg body weight) ameliorated serum ALT levels in those rats fed a Zn-deficiency diet [88]. Interestingly, the Se supplementation in that study modulated the Zn level in the experimental model. The Se treatment may initiate an increase in insulin activity, and may subsequently improve Zn deficiency in the experimental model. In fact, we found a positive correlation between the serum Zn and Se levels in patients with HCV-related CLD [34]. Another study revealed that the Se-enriched *lactobacillus* reversed CCl_4_-induced liver injury by facilitating antioxidant enzyme activity and inhibiting lipid peroxidase activity [87]. Unfortunately, beneficial effects of antioxidants, including Se, ascorbic acid, and α-tocopherol, on serum transaminase levels and HCV RNA load were not apparent in a clinical trial [78].

#### 2.2.3. Fe 

It is well established that Fe deposition in the liver can initiate ROS and subsequently lead to hepatic inflammation, lipid peroxidation, and insulin resistance [58]. The Fe overload is commonly observed in patients with HCV-related CLD or those with NAFLD, which is characterized by excessive hepatic fat accumulation and no history of alcohol abuse [60] (Table 2) as well as those with hereditary hemochromatosis [89]. We confirmed that serum ferritin levels were closely associated with ALT levels in patients with HCV-related CLD, indicating that the Fe deposit in the liver might evoke hepatic inflammation [48]. In patients with chronic hepatitis C, attenuation of the Fe overload by phlebotomy, which is usually performed as a treatment for hemochromatosis, eventually caused the decline of serum transaminase levels [79,80] as well as those with NAFLD [81,82] (Table 3).

Iron regulatory factors, such as hepcidin, ferroportin, and transferrin receptor, also play crucial roles in iron homeostasis. Thus, dysregulation of these factors can initiate Fe storage in the liver. Hepcidin is a peptide hormone produced by the liver that binds to ferroportin and inhibits Fe absorption from the small intestine [90]. The expression of hepcidin was suppressed by HCV-induced oxidative stress in an in vitro study [62]. In contrast, serum hepcidin levels were elevated in patients with NAFLD [63] (Table 2).

#### 2.2.4. Cu

Cu is also another essential trace element that participates in many enzymatic and redox reactions. The circulating Cu concentration was frequently elevated in patients with chronic hepatitis C [33,44,46,50,64] (Table 2). Increased Cu levels may be derived from the facilitation of ceruloplasmin synthesis by interleukin (IL)-1 in Cu deficient rats [91]. Ceruloplasmin, a transporter for Cu, is a glycoprotein which is synthesized in the liver, and plays an essential role in the acute-phase reaction [92]. Indeed, serum ceruloplasmin levels were significantly associated with ballooning hepatocytes, inflammatory cells infiltration, and/or hepatic steatosis in pediatric NAFLD patients [66].

The elevated Cu and ceruloplasmin levels may contribute to the inflammatory change in the liver of patients with chronic hepatitis. The Cu-induced MT also may account for the increase of serum Cu level in patients with chronic hepatitis C. Excessive Cu is considered to be related to the induction of MT. The Cu-induced MT probably initiates hydroxyl radicals and subsequently leads to the inflammation in the liver [93]. 

However, the co-administration of Cu and imatinib mesylate, a tyrosine kinase inhibitor, exhibited the anti-inflammatory and anti-fibrotic effects in HCC-induced rats [94]. These effects may be derived from an anti-inflammatory action of Cu by suppressing the activity of the protein complex, nuclear factor kappa-light-chain-enhancer of activated B cells (NF-κB).

### 2.3. The Roles of Trace Elements in Hepatic Fibrosis 

#### 2.3.1. Zn

Zn directly inhibits the fibrotic process in the liver via the actions of metalloproteinase (MMP) or prolyl-hydroxylase [95,96] (Figure 1). Zn also inhibits fibrosis via its anti-inflammatory, anti-apoptotic, and antioxidant properties, or by controlling the function of hepatic stellate cells (HSCs) [97]. Zn deficiency thus participates in hepatic fibrosis both directly and indirectly. 

An interesting study documented the improvement of hepatic fibrosis by administration of polaprezinc in vitro [98] and in vivo [99] studies. Polaprezinc supplementation (2.2 g/kg weight) led to the improvement of hepatic fibrosis in an animal NASH model by promoting fibrolysis via the inhibition of the tissue inhibitor of metalloproteinase (TIMP)-1 activity and by reducing the activity of HSCs [99]. Kono et al. also confirmed the inhibitory effects of polaprezinc (200 mg/kg weight/day) on hepatic fibrosis in a rat model of thioacetamide-induced hepatic fibrosis [100]. The effect of zinc oxide nanoparticle (10 mg/kg weight/day) on hepatic fibrosis was also demonstrated in rats [101].

A clinical trial revealed that the administration of polaprezinc (150 mg/day for six months) reduced the type IV collagen levels and the TIMP-1 activity in the sera of patients with liver cirrhosis [74] (Table 3). Another study documented that a low dose of zinc sulfate (50 mg Zn/day) can prevent the deterioration of clinical status and suppress excessive Cu accumulation in non-alcoholic cirrhotic patients [75]. On the contrary, a larger amount of Zn administration results in severe Cu deficiency [102].

#### 2.3.2. Se

A decline in the serum Se level is frequently observed in patients with liver cirrhosis [103,104], because Se is transported into the blood by binding to Se-containing proteins such as albumin and selenoprotein P [83]. However, the decrease in the serum Se level indicates hepatic dysfunction rather than Se deficiency in such patients [103]. We demonstrated that serum Se levels were reduced in proportion to the severity of hepatic fibrosis in patients with an HCV-related CLD [34]. In another study, the circulating and hepatic Se concentrations were markedly decreased in N-nitrosodimethylamine-induced hepatic fibrosis [105]. In this animal model of hepatic fibrosis, decreases in Se and GPx levels may participate in the impairment of an antioxidant defense, and may trigger the process of hepatic fibrosis. 

Ding et al. elucidated that the administration of sodium selenite (200 μg/kg weight body diet) inhibited hepatic fibrosis in mice by suppressing the number of collagen producing HSCs and by promoting collagen degradation [106]. In the clinical trial, the efficacy of selenite (200 or 400 μg/day) was revealed in patients with liver cirrhosis [76]. Further investigations should be required to clarify the inhibitory effect of Se supplementation against hepatic fibrosis in a clinical trial.

#### 2.3.3. Fe

The Fe overload facilitates hepatic fibrosis in patients with HCV-related CLD, NASH, alcoholic liver disease, or hemochromatosis [107]. Excessive Fe can initiate the Fenton reaction to generate a large amount of free radicals, subsequently leading to tissue damage in the liver, and finally contributing to hepatic fibrosis. In addition, excessive Fe can promote the signals for fibrosis in parenchymal and non-parenchymal cells.

Several studies have demonstrated that the hepatic Fe accumulation was associated with the severity of hepatic fibrosis in patients with HCV-related CLD [108,109]. Phlebotomy resulted in the improvement of the hepatic fibrosis as well as hepatic inflammation in patients with chronic hepatitis C [80] (Table 3). Likewise, Fe deposits in the liver may reflect the degree of hepatic fibrosis in patients with NAFLD. Indeed, an increase in the serum ferritin level is closely associated with hepatic Fe deposition, suggesting that the serum ferritin level can be a surrogate marker for hepatic Fe deposition [107]. Therefore, higher serum ferritin levels could be used to predict more advanced fibrosis in such patients [61]. In contrast, Fe depletion by phlebotomy appeared to ameliorate hepatic fibrosis in patients with NAFLD [82] (Table 3).

The serum ferritin level can predict even early mortality in patients with decompensated liver cirrhosis [110]. Similarly, the serum ferritin levels of patients with liver cirrhosis were further increased as the patients’ hepatic reserve deteriorated severely [111].

#### 2.3.4. Cu

The circulating Cu concentration is also elevated in proportion to the severity of hepatic fibrosis [112]. Cu acts as a cofactor against hepatic fibrosis in chronic liver disease, particularly in the biosynthesis of collagen.

The Cu/Zn ratio was markedly elevated in patients with liver cirrhosis, compared to that of patients with chronic hepatitis or normal healthy controls [113,114,115]. It is of interest that a higher Cu/Zn ratio was associated with mortality in those patients [113]. 

In male HBV-related cirrhotic patients, serum ceruloplasmin levels were inversely correlated with hepatic fibrosis, because ceruloplasmin is synthesized in the liver [116]. Likewise, HCV-related cirrhotic patients with hepatic encephalopathy had lower ceruloplasmin concentrations in their sera than those without hepatic encephalopathy and normal healthy subjects [117]. 

### 2.4. The Roles of Trace Elements in Hepatic Steatosis

#### 2.4.1. Zn 

Zn is involved in the activation of peroxisome proliferator-activated receptor-α (PPAR-α), a regulator of lipid homeostasis [118]. Zn may participate in the DNA-binding activity of PPAR-α. Therefore, Zn deficiency may result in a reduction of PPAR-α activity, and consequently the promotion of lipid peroxidation, finally leading to the deterioration of hepatic steatosis (Figure 1).

Indeed, compared to normal control rats, the serum Zn levels and hepatic Zn contents were significantly lower in an experimental animal model of fatty liver, which was induced by tetracyclin [119]. We verified the correlation between Zn deficiency and hepatic steatosis in patients with HCV-related CLD, because hepatic steatosis is one of the common histological features in the liver of those patients [120]. The serum Zn levels of patients gradually decreased as their hepatic steatosis developed from a mild status to a severe status, based on the criteria for hepatic steatosis proposed by Brunt et al. [121], for patients with HCV-related CLD [59]. Approximately equivalent results were obtained from the study by Guo et al.: Serum Zn levels were significantly lower in patients with both chronic hepatitis C and NAFLD compared to patients with chronic hepatitis C alone [50]. 

It is also well-known that insulin resistance contributes to the process of hepatic steatosis in patients with HCV-related CLD [122]. Our previous study confirmed a close correlation between the severity of hepatic steatosis and the value of a homeostasis model for assessment of insulin resistance (HOMA-IR), an indicator of insulin resistance in such patients [59]. Moreover, our previous studies revealed that serum Zn levels were inversely correlated with the values of HOMA-IR, suggesting that Zn deficiency results in the development of insulin resistance in patients with HCV-related CLD [48,59].

Likewise, a recent report by Asprouli et al. documented that serum Zn levels were also decreased in proportion to the grade of hepatic steatosis in NAFLD patients [52]. Serum Zn levels were inversely correlated with HOMA-IR values in those patients [53]. A lower dietary intake of Zn may account for Zn deficiency in NAFLD patients [123]. It is of interest that the administration of zinc sulfate reversed ethanol-induced hepatic steatosis in mice by reactivating PPAR-α and hepatocyte nuclear factor-4α [124]. 

#### 2.4.2. Se

The serum Se levels of patients with both chronic hepatitis C and NAFLD were also lower than those of patients with chronic hepatitis C alone [50]. However, the serum Se levels of NAFLD patients gradually rose as their hepatic steatosis progressed to a severe status [56] (Table 2). Serum GPx levels were also elevated in those patients [57]. The putative mechanism by which Se contributes to the progression of hepatic steatosis in NAFLD has not been fully established. Insulin resistance via the activation of a selenoprotein may account for these phenomena in NAFLD patients. In fact, the serum Se levels were positively correlated with the values of HOMA-IR in patients with NAFLD patients [56]. Misu et al. elucidated the positive correlation between selenoprotein P mRNA levels and the severity of insulin resistance in patients with type 2 diabetes mellitus [125]. Further analyses are necessary to clarify the relationship between serum Se levels and hepatic steatosis in NAFLD patients.

#### 2.4.3. Fe 

When the transgenic mice expressing the HCV polyprotein were fed an excessive Fe diet, the development of hepatic steatosis was observed through activation of the unfold protein [126]. Our previous study confirmed that serum ferritin levels were increased in proportion to the grade of hepatic steatosis in patients with HCV-related CLD [59]. In the experimental animal model of NAFLD, the hepatic Fe overload was induced prior to the development of hepatic steatosis and insulin resistance [127]. The association of a hepatic Fe deposit with hepatic steatosis was also shown in patients with NAFLD [61]. Indeed, phlebotomy turned out to attenuate hepatic steatosis in those patients [80,81] (Table 3).

#### 2.4.4. Cu

The serum Cu levels of individuals with NAFLD were often decreased [128] (Table 2). Inadequate Cu availability is likely to increase lipid accumulation in the liver. Thus, lower Cu bioavailability may affect lipid metabolism and it may be involved in the development of NAFLD [65]. Moreover, Cu deficiency can initiate the alteration of mitochondrial morphology, leading to an impairment of fatty acid β oxidation. These phenomena can affect hepatic lipid accumulation in patients with NAFLD (Figure 2).

It is noteworthy that circulating ceruloplasmin levels were also reduced in NAFLD patients [66]. However, chronic hepatitis B patients with hepatic steatosis had higher ceruloplasmin levels in their sera than those without steatosis. Moreover, serum ceruloplasmin levels were associated with the severity of hepatic steatosis in patients with chronic hepatitis B [129].

Cu seems to play an important role in the Fe homeostasis and be associated with the Fe perturbation observed in NAFLD [63]. Cu is required for the function of ceruloplasmin to export Fe from the liver or the reticuloendothelial system, and for the expression of ferroportin. The membrane-bound type of ceruloplasmin might be mandatory for the stability of ferroportin. Consequently, a lower hepatic Cu content and a lower serum Cu concentration eventually cause the Fe overload in patients with NAFLD. 

A high fructose diet and Cu restriction may trigger hepatic steatosis and inflammation in mice [130], because dietary fructose inhibits the duodenal Cu absorption by suppressing duodenal expression of copper transporter-1 (ctr-1) [131], which may be the primary protein responsible for the import of dietary Cu.

Hepatic steatosis is frequently observed in patients with Wilson’s disease [132], which is the autosomal recessive hereditary disease, and is caused by the gene mutation of ATP7B essential in the Cu metabolism [133]. This mutation results in an impaired hepatic Cu excretion and subsequently Cu accumulation in the liver. Liggi et al. elucidated the close correlation between the serum Cu level and grading of hepatic steatosis in those patients, indicating that hepatic steatosis in Wilson’s disease may not be derived from metabolic comorbidities but from Cu accumulation in the liver [134].

### 2.5. Roles of Trace Elements in Autoimmune Liver Diseases

#### 2.5.1. AIH

Several research groups explored the serum ferritin levels in patients with autoimmune hepatitis (AIH). AIH is largely a chronic necroinflammatory disease of the liver, characterized by hypergammaglobulinemia and circulating autoantibodies [135]. According to a recent study, hyperferritinemia was commonly observed in AIH patients at baseline [136]. Ferritin is an acute-phase reactant, and an increase in the serum ferritin level may reflect the promotion of pro-inflammatory cytokines such as IL-6 in such patients [137]. The serum ferritin levels were thus significantly correlated with serum ALT levels in those patients. The study’s authors also revealed that higher levels of serum ferritin and lower levels of serum IgG at baseline could predict more favorable responses to immunosuppressive treatments [136]. Notably, hyperferritinemia in AIH patients with a biochemical response seemed to be independent of the serum hepcidin level. In addition, the severity of Fe deposit in the liver did not affect a response to the treatment at all [136]. 

Another study revealed that serum ferritin levels were lower in patients with AIH than in patients with chronic hepatitis B, and that the serum hepcidin levels were far lower in patients with AIH than in patients with chronic hepatitis B [138]. Hepcidin is recognized as a chemotactic factor for T-lymphocytes, dendritic cells, monocytes, and mast cells. A decrease in the synthesis of hepcidin may cause an imperfect interplay between the innate and adaptive immune systems in patients with AIH.

Declines in the serum Zn [139] and Se [140] levels of patients with AIH have been reported. Indeed, lower serum Zn and Se levels were one of the common features in autoimmune diseases [141,142]. However, lower serum Zn and Se levels in AIH patients may reflect a consequence of chronic liver damage rather than an autoimmune phenomenon. 

#### 2.5.2. PBC

Primary biliary cholangitis (PBC) is an autoimmune cholestatic liver disease characterized by nonsuppurative inflammatory destruction of the interlobular bile ducts [143]. It is well established that cholestasis causes Cu accumulation in the liver. Orcein stain, which indicates the existence of copper-associated protein, was frequently useful for the diagnosis of PBC [144]. PBC patients positive for the orcein stain in the liver specimens had higher total bilirubin and alkaline phosphatase levels than those negative for the orcein stain [145]. In addition, the serum Zn level of PBC patients gradually decreased as the clinical stage became more severe [146].

The serum Se level was also decreased in patients with PBC [147]. The decline in the serum Se level in PBC patients may be the consequence of chronic liver damage rather than Se deficiency [103]. Indeed, Se supplementation did not affect the liver function of PBC patients [77] (Table 3). 

### 2.6. The Role of Genetic Polymorphism in the Trace Elements

A genome-wide association (GWA) study identified a single nucleotide polymorphism (SNP) in rs738409 in the patatin-like phospholipase domain containing 3 (PNPLA3) gene, which is recognized as an adiponutrin gene, and the SNP was strongly associated with the grade of hepatic fat content [148]. The PNPLA3 risk allele homozygosity was associated with an increased risk for NASH [149]. Moreover, rs738409 also showed a strong association with hepatic iron deposition in Japanese NAFLD patients [150]. 

The PNPLA3 G allele also affects the development of hepatic steatosis in Wilson’s disease. However, the PNPLA3 G allele was not associated with the hepatic Cu content in such patients [151].

The IL-6-174 G/C promotor polymorphism affected the hepatic Zn content in autopsy cases [152]. Since Zn supplementation (9.08 mg/100 g weight of zinc sulfate) ameliorates the synthesis of IL-6 in salmonella-induced hepatic damage in a murine model [153], an IL-6 SNP may be involved in the pathogenesis of Zn deficiency in patients with CLDs.

### 2.7. The Roles of Microbiota in Trace Elements

Sequencing the 16S gene ribosomal RNA (rRNA) has become a popular method for identifying bacterial communities [154]. Next-generation sequencing enabled us to investigate the relationship between the composition of microbiomes and dysbiosis [155].

Gut dysbiosis frequently causes a variety of critical complications, including endotoxemia and hepatic encephalopathy, in patients with liver cirrhosis. These severe complications are likely to be derived from small intestinal bacterial overgrowth and/or an increase in intestinal permeability, termed “leaky gut” [156]. Notably, bacterial translocation is ordinarily observed in the portal vein as well as hepatic and peripheral blood of patients with decompensated liver cirrhosis [157]. Bacterial translocation of gut flora was dominated primarily by the *Proteobacteria phylum* in such patients [158]. These species are commonly present in both the peripheral and portal blood of patients with liver cirrhosis, and were considered to be strong producers of iron chelates (siderophores). Indeed, both the peripheral blood and portal blood of cirrhotic patients were enriched in bacterial Kegg Orthologous genes linked to active Fe transport [159]. These results may suggest that *Proteobacteria phylum* is functionally linked to Fe metabolism.

It is well recognized that the microbiota also plays a crucial role in the pathogenesis of NAFLD [160]. Song et al. documented that (1) a dietary Cu-fructose interaction regulated gut microbiota and that (2) the alteration of gut flora via the gut barrier dysfunction might result in the development of steatosis in the liver. The gut barrier dysfunction was derived from markedly downregulated intestinal tight junction proteins and increased gut permeability [161]. Indeed, the alterations of microbiome indicated an increase in Firmicutes and a depletion of *Akkermansia*, which is considered to be crucial for maintaining the gut burrier function. 

### 2.8. The Roles of Sarcopenia in Trace Elements

Sarcopenia, characterized by a loss of skeletal muscle mass and low muscle strength [162], is one of the common features in patients with CLD, which is observed even in NAFLD patients [163] as well as cirrhosis patients [164]. It is well documented that several types of minerals, including Mg, calcium (Ca), and Se, play pivotal roles in muscle metabolism [7,165]. It remains controversial whether an excessive Fe status contributes to sarcopenia [166,167]. Nishikawa et al. contended that Zn deficiency might account for sarcopenia in patients with CLD [168], although a putative mechanism by which Zn deficiency causes sarcopenia in such patients remains unclear. The combination treatment of Zn with a branched chain amino acid may be effective in liver cirrhosis patients with sarcopenia [169]. 

Several studies revealed that in elderly people, lower serum Se levels are associated with a lower muscle mass [170]. It is plausible that treatment with fish oil and Se attenuated skeletal muscle atrophy by preventing a rise in myostatin [171], which is a negative regulator of muscle mass [172].

### 2.9. The Role of MicroRNA in Trace Elements

MicroRNAs (miRNAs), which are small, single stranded non-coding RNAs of 19–25 nucleotide in length, negatively regulate gene expression via translational inhibition or messenger RNA (mRNA) degradation [173]. Many miRNAs play essential roles in diverse biological processes, including cell differentiation, proliferation, migration, and survival [174].

Some kinds of miRNAs are also involved in the pathogenesis of NAFLD [175]. Especially, increases in the expressions of miR-200a and miR-223 were negatively correlated with iron regulatory protein 1 (IRP1) in a mouse model of NAFLD, implying that miRNAs might contribute to Fe homeostasis in patients with NAFLD [176]. 

## 3. Conclusions

Much novel evidence has accumulated regarding the roles of essential trace elements in CLDs by recent advances in various types of molecular biological technologies. Some of the trace elements were extremely useful for the prediction of the prognosis in patients with CLDs. The beneficial therapeutic effects of some trace elements supplementations have been confirmed in experimental animal models and/or clinical trials. However, the evidence levels remain relatively low. Further prospective multicenter cohort studies should be conducted to investigate the usefulness of the essential trace elements in CLDs.

## Figures and Tables

**Figure 1 nutrients-12-02084-f001:**
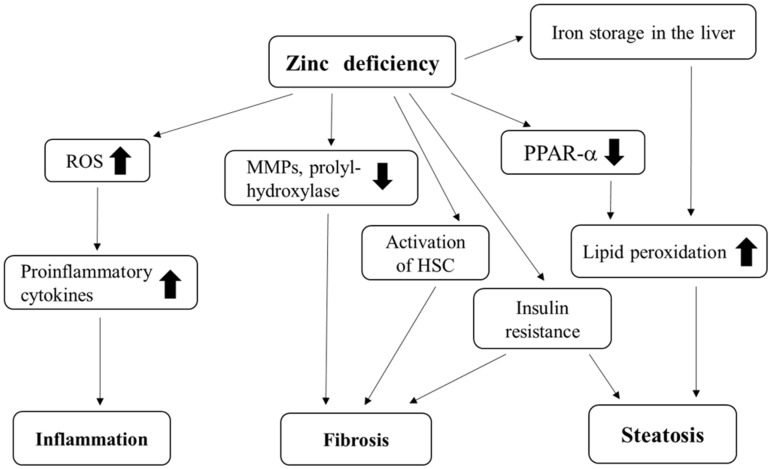
Relationship between zinc deficiency and inflammation, fibrosis or steatosis in the liver. ROS: Reactive oxygen species; MMP: Metalloproteinase; PPARα: Peroxisome proliferator-activated receptor-α; HSC: Hepatic stellate cell; ⬆: Promotion; ⬇: Inhibition.

**Figure 2 nutrients-12-02084-f002:**
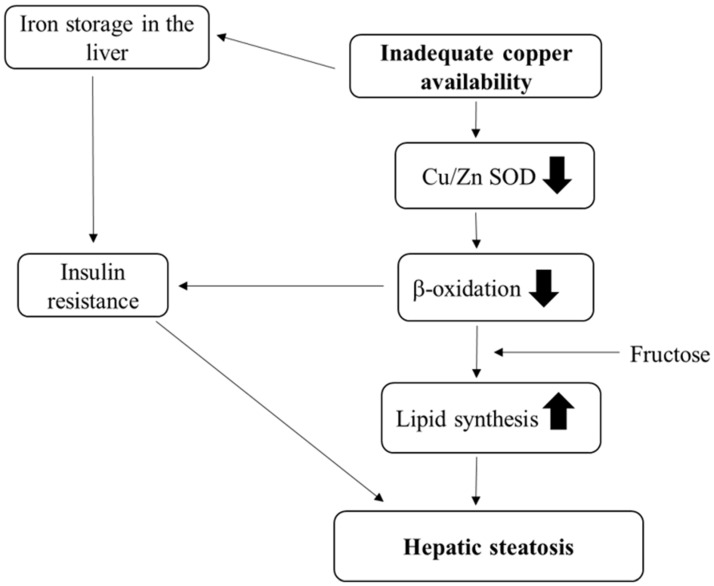
Putative mechanisms by which inadequate copper availability causes hepatic steatosis. SOD: Superoxide dismutase; ⬆: Promotion; ⬇: Inhibition.

**Table 1 nutrients-12-02084-t001:** Roles of trace elements in the translation, transcription, and replication of hepatitis viruses.

Hepatitis Viruses	Trace Elements	Function of Trace Element	References
HBV	Zn	Zn deficiency caused poor response to HBV vaccination	[10,11]
	Se	Sodium selenite suppressed HBV protein expression, transcription, and genome replication	[11]
HCV	Zn	negative regulator of HCV replication	[12,13]
		initiation of IFN-α	[14]
	Se	intracellular replication of HCV	[15]
		initiation of selenoprotein P by HCV infection	[16]
	Fe	promotion/inhibition of HCV replication (controversial)	[17,18]
		promotion of HCV translation	[19,20]
	Cu	Cuprous oxide inhibited the entry of HCV pseudoparticle	[21]
HEV	Zn	Zinc sulfate and zinc acetate inhibited the activity of viral RNA-dependent RNA polymerase	[22]

HBV: Hepatitis B virus; HCV: Hepatitis C virus; HEV: Hepatitis E virus; Zn: Zinc; Se: Selenium; Fe: Iron; Cu: Copper; IFN: Interferon.

**Table 2 nutrients-12-02084-t002:** Comparisons of the status of trace elements between HCV-related CLD and NAFLD/NASH.

Trace Elements	HCV-Related CLD	NAFLD/NASH
Zn	low	low
	[46,47,48,49,50]	[51,52,53]
Se or GPx	low	high
	[33,34,54,55]	[56,57]
Fe (ferritin)	high	high
	[48,58,59]	[60,61]
hepcidin	low	high
	[62]	[63]
Cu	high	low
	[33,46,64]	[65]
ceruloplasmin	unknown	low
		[66]

HCV: Hepatitis C virus; NAFLD: Nonalcoholic fatty liver disease; NASH: Nonalcoholic steatohepatitis; Zn: Zinc; Se: Selenium; Fe: Iron; Cu: Copper.

**Table 3 nutrients-12-02084-t003:** Summary of the clinical trials on administration or depletion of the trace elements in patients with CLDs.

Trace Elements	Formulations and Dosages	Assigned Patients	Effects by Trace Elements	References
Zn	polaprezinc, 225 mg	chronic hepatitis C	improvement of serum ALT level	[67,68]
			improvement of serum ferritin level	[67]
	polaprezinc, 150 mg	liver cirrhosis	attenuation of hepatic fibrosis	[74]
	zinc sulfate, 50 mg		prevention of present clinical status deterioration	[75]
IFN-based treatment combined with Zn	polaprezinc, 150 mg	chronic hepatitis C	higher CR rate than the treatment with IFN alone	[69]
			lower ALT levels than the treatment with IFN alone	[70]
			lower incident of gastrointestinal adverse effects	[71]
Se	selenite, 200 or 400 µg	liver cirrhosis	improvement of hepatic reserve	[76]
	200 µg selenium	PBC	insignificant	[77]
combined treatment of ascorbic acid and α-tocophenol with Se	200 µg selenium	chronic hepatitis C	insignificant	[78]
Fe depletion (phlebotomy)		chronic hepatitis C	improvement of serum ALT level	[79,80]
			improvement of serum ferritin level	[79,80]
			improvement of hepatic fibrosis	[80]
		NAFLD	improvement of serum ALT level	[81,82]
			improvement of serum ferritin level	[81,82]
			improvement of hepatic fibrosis	[81]
			improvement of hepatic steatosis	[81,82]

Zn: Zinc; Se: Selenium; Fe: Iron; IFN: Interferon; PBC: Primary biliary cholangitis; NAFLD: Nonalcoholic fatty liver disease; ALT: Alanine aminotransferase; CR: Complete response.

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
