# Peer review of "Current Trends of Essential Trace Elements in Patients with Chronic Liver Diseases"

_nutrients, 2020, doi:10.3390/nu12072084_

Round 1

Reviewer 1 Report

The authors submit a review aiming at gathering the knowledge about the relations that have been identified between trace elements and liver disease focusing on chronic liver diseases, including chronic hepatitis, liver cirrhosis, nonalcoholic fatty liver disease, and autoimmune liver diseases. Correctly they state that deficiency or an excess of some trace elements may cause these metabolic disorders - which is not new as the 11 essential trace elements are involved in all steps of metabolism.

Although such a review may be useful, the authors miss the objective due to lack of concision and practicalities - many statements are very vague - never a dose is mentioned - which is cornerstone to interventions - this information should be retrieved from the papers that are cited.

The paper is too long, and contains digression about pathology that have nothing to do with trace elements. Multiple references, which should be drastically reduced, are oriented on liver pathology and not on trace elements. Finally the authors are not English natives, which results, not in faults, but in multiple imprecise and awkward formulations.  The text reads as an association of multiples sentences that do not become a text: very difficult to read.

Abstract: please go to the point, mention the trace elements that will be discussed. There are not data and it does not reflect the paper.

Line 29: what do you mean with many “types” of trace elements. Metal? Metalloids? Essentila? Non essential? Either there is a language problem through the text, or a lack of basic knowledge regarding trace elements.

Lie 31: delete “S” after function

Line 42: here again “types” - what do you mean

Lines 46, 58, 75: as the title of the section is already the “roles of trace elements” … please delete the 4 words from the following subtitles, it unnecessarily adds words.

Line 48: “Some types” ?? what does it mean

Line 53: please reformulate to “Table 1 summarizes trace element functions

The word “role” is also used in a way that bring confusion: does is stand for “function”, or?

Line 61: replace “can” by “may”

Line 64: what is the responsiveness to vaccination? Vaccination is repeated multiple times int eh papragraph. please reformulate the § lines 65-69

Line 73: Se is “likely” to… this formulation is not adequate

Line 78-9: HCV infection is thought initiate a tremendous level?  Please reformulate, not clear

Line 93: genes are knocked out - not down.

Lines 101-108: lots of imprecisions render this § a problem. The decline is systemic concentrations of Se are clearly related to inflammation (1), the mechanisms mentioned in the paragraph are presented in a very unclear way and can only be additional  (1. Duncan A, Talwar D, McMillan DC, Stefanowicz F, O'Reilly DS. Quantitative data on the magnitude of the systemic inflammatory response and its effect on micronutrient status based on plasma measurements. Am J Clin Nutr 2012;95:64-71.)

Lines 131-152-168-185: here again as the tile of section 2.2. contains the roles of TE in hepatic inflammation, this should be deleted from the subsequent subtitles - just indicate the name of the trace element (zinc, selenium… ) you refer to in the specific subsection

Lines 204-229-241-259: same problem for that section - delete main title name - it is just an irritating, useless repetition.

Lines 277-304-317-326: same issue

Line 134: zinc deficiency: a low plasma value does not necessarily mean deficiency, but response to inflammation most of the time - please be more precise

Table 2:

delete “references” in the table - anyone can understand that this is reference and reduces the visibility of content

Why is not GPX mentioned in this table? If you include hepcidin and caeruloplasmin it must be present

Line 141..: if you discuss zinc supplementation a dose is required.

Table 3: doses of Zn and Se used in the studies are badly required

Lines 153-167: this § is problematic as the authors do not seem to know that selenium is not an antioxidant by itself but only when if becomes incorporated into a selenopeptide. The most important, Se-Cys is never mentioned nor is the main antioxidant enzyme family Glutathione peroxidase (GPX). Searching PubMed for “GPX and liver retrieves 2930 hits….).

Line 169: verb is missing

Line 204…: multiple unnecessary repetitions in the paragraph.

Lines 219 on: please provide doses.

Lines 308: selenium deficiency or excess - not clear

Line 312-3: verb is missing

Line 333: pleas reformulate “A diet high fructose feeding” to a high fructose diet

Line 341-2: do not repeat 2 time Wilsons disease in same sentence.

Lines 399-421: the complete section 2.7 should be deleted as unrelated to the topic and containing not one mention about any of the 4 trace elements

Line 434: “reasonable” - please reformulate445: please delete as only very indrectl related to the topic.

Lines 437-

Reference style in the text:  question to the editors: should the authors use as thy do “.. and colleagues” or the shorter more efficient “.. et al”?

Author Response

Reviewer#1

The authors submit a review aiming at gathering the knowledge about the relations that have been identified between trace elements and liver disease focusing on chronic liver diseases, including chronic hepatitis, liver cirrhosis, nonalcoholic fatty liver disease, and autoimmune liver diseases. Correctly they state that deficiency or an excess of some trace elements may cause these metabolic disorders - which is not new as the 11 essential trace elements are involved in all steps of metabolism.

(Response)

Of course, it is well known that some trace elements may cause these metabolic disorders. But, this review article focuses on the novel function of trace elements in chronic liver diseases.

Although such a review may be useful, the authors miss the objective due to lack of concision and practicalities - many statements are very vague - never a dose is mentioned - which is cornerstone to interventions - this information should be retrieved from the papers that are cited.

(Response)

Thank you for giving us an important comment. The dosages of Zn and Se supplementation were retrieved from the references (see the Text, red highlight).

The paper is too long, and contains digression about pathology that have nothing to do with trace elements. Multiple references, which should be drastically reduced, are oriented on liver pathology and not on trace elements. Finally the authors are not English natives, which results, not in faults, but in multiple imprecise and awkward formulations.  The text reads as an association of multiples sentences that do not become a text: very difficult to read.

(Response)

I tried to simplify our article. But, the article does not seem to contain digression at all. So, any parts cannot be deleted. Our manuscript has been already checked by a reliable native speaker before submitting. But, there are some parts to be corrected grammatically in the manuscript. Grammatical errors which I found were corrected.

Abstract: please go to the point, mention the trace elements that will be discussed. There are not data and it does not reflect the paper.

(Response)

In accordance with a reviewer’s comment, the results were inserted in “Abstract” as shown below.

A genome-wide association study revealed that a peculiar genetic polymorphism affected the metabolism of a trace element. The impairment of a trace element homeostasis caused gut dysbiosis (see “Abstract”, red highlight).

Line 29: what do you mean with many “types” of trace elements. Metal? Metalloids? Essentila? Non essential? Either there is a language problem through the text, or a lack of basic knowledge regarding trace elements.

Line 42: here again “types” - what do you mean?

Line 48: “Some types” ?? what does it mean

(Response)

“Type” was deleted in the text.(see the text)

Line 31: delete “S” after function

(Response) “functions” was corrected to “function”.(see the Text)

Lines 46, 58, 75: as the title of the section is already the “roles of trace elements” … please delete the 4 words from the following subtitles, it unnecessarily adds words.

(Response)

In accordance with a reviewer’s comment, the 4 words were deleted.(see the Text)

Line 53: please reformulate to “Table 1 summarizes trace element functions

The word “role” is also used in a way that bring confusion: does is stand for “function”, or?

(Response)

In accordance with reviewer’s comments, Table 1 was revised (see Table 1)

“Roles” were replaced with “function” in Table 1.

Line 61: replace “can” by “may”

(response) “Can was replaced with “may.

Line 64: what is the responsiveness to vaccination? Vaccination is repeated multiple times int eh papragraph. please reformulate the § lines 65-69

(Response)

Thank you for an appropriate comment. In accordance with a reviewer’s comment, the statement was revised as shown below.

. The responsiveness was evaluated by serum anti-HBs level. The serum anti-HBs level was markedly decreased in rats fed a diet with lower Zn content [12].(see theText, red highlight).

Line 73: Se is “likely” to… this formulation is not adequate

(Response)

In accordance with a reviewer’s comment, the statement was revised as shown below.

Se is likely to activate p53 by promoting its expression and phosphorylating multiple sites, and suppress the activities of HBV promoters and enhancers. (see the text, red highlight).

Line 78-9: HCV infection is thought initiate a tremendous level?  Please reformulate, not clear

(Response)

I apologize for an unclear statement. The statement was corrected as shown below.

Chronic HCV infection is thought to cause the production of reactive oxygen species (ROC) and subsequently inflammation and fibrosis in the liver, leading to chronic liver damage, including chronic hepatitis, liver cirrhosis and ultimately to HCC [16].(see the Text, redhighlight)

Line 93: genes are knocked out - not down.

(Response) “Knock down” was replaced with “knock out”.(see the text, redhighlight)

Lines 101-108: lots of imprecisions render this § a problem. The decline is systemic concentrations of Se are clearly related to inflammation (1), the mechanisms mentioned in the paragraph are presented in a very unclear way and can only be additional  (1. Duncan A, Talwar D, McMillan DC, Stefanowicz F, O'Reilly DS. Quantitative data on the magnitude of the systemic inflammatory response and its effect on micronutrient status based on plasma measurements. Am J Clin Nutr 2012;95:64-71.)

(Response)

In accordance with a reviewer’s comment, the revised statements were shown. The reference written by Duncan et al was cited in this paragraph.(see the Text, red highlight)

A decline in the systemic Se concentration may be attributable to an intracellular replication of HCV. RNA viruses, including HCV and human immunodeficiency virus (HIV), encode Se-dependent glutathione peroxidase (GPx) module [27], which is one of selenoproteins and protects against damage induced by free radicals. It is plausible that serum Se levels were negatively correlated with the loads of HCV RNA in patients with chronic hepatitis C [28], but we did not confirm this phenomenon in patients with HCV-related CLD [29]. The decrease in circulating Se concentration may also reflect a systemic inflammatory response [30].Notably, Murai et al. demonstrated that hepatic selenoprotein P mRNA was upregulated by HCV infection, and that its knockout in hepatocytes caused an induction of IFN-stimulated genes and a subsequently inhibited the replication of HCV.

Lines 131-152-168-185: here again as the tile of section 2.2. contains the roles of TE in hepatic inflammation, this should be deleted from the subsequent subtitles - just indicate the name of the trace element (zinc, selenium… ) you refer to in the specific subsection

Lines 204-229-241-259: same problem for that section - delete main title name - it is just an irritating, useless repetition.

Lines 277-304-317-326: same issue

(Response)

The subtitles were revised on the basis of a reviewer’s comment, (see the Text, red highlight).

Line 134: zinc deficiency: a low plasma value does not necessarily mean deficiency, but response to inflammation most of the time - please be more precise

(Response)

In accordance with a reviewer’s comment, the statement shown below (red highlight) was inserted in this paragraph.

Zn deficiency thus causes the production of ROS, and subsequently leads to inflammation in the liver (Figure 1). Zn deficiency is often observed in patients with chronic hepatitis [27, 44-51] or in those with NAFLD [52] (Table 2), although lower Zn concentration does not necessarily reflect the inflammation process [2]. The serum Zn levels of patients with chronic hepatitis were inversely correlated with serum transaminase levels [48] and with their histological activity scores [46]. (See the text).

Table 2:

delete “references” in the table - anyone can understand that this is reference and reduces the visibility of content

Why is not GPX mentioned in this table? If you include hepcidin and caeruloplasmin it must be present

(Response)

In accordance with a reviewer’s comments, “references” were deleted in Table 2.

Also, the statements on GPx were inserted in Table 2.

Line 141..: if you discuss zinc supplementation a dose is required.

Table 3: doses of Zn and Se used in the studies are badly required

Lines 219 on: please provide doses.

(Response)

Dosages of Zn and Se supplementation were addressed in the text and Table 3 (red highlight).

Lines 153-167: this § is problematic as the authors do not seem to know that selenium is not an antioxidant by itself but only when if becomes incorporated into a selenopeptide. The most important, Se-Cys is never mentioned nor is the main antioxidant enzyme family Glutathione peroxidase (GPX). Searching PubMed for “GPX and liver retrieves 2930 hits….).

(Response)

We appreciate an appropriate comment by a reviewer. The1st paragraph in 2.2.2. (Se) was revised as shown below (see the Text, red highlight).

Se is a constituent of GPx which protects against the damage induced by ROS [62]. Reduced serum Se levels are frequently observed in patients with chronic hepatitis [28, 29, 47, 53, 63, 64]. Serum GPx levels were also lower in patients with chronic hepatitis C than those in cases of normal healthy control [29,65, 66] (Table 2).

Line 169: verb is missing

(Response)

The sentence shown below is grammatically correct. A verb is not missing in the sentence. It does not need to be revised.

.It is well established that Fe deposition in the liver can initiate reactive oxygen species (ROS) and subsequently lead to hepatic inflammation, lipid peroxidation and insulin resistance.

Line 312-3: verb is missing

(Response)

The sentence shown below is grammatically correct. A verb is not missing in the sentence. It does not need to be revised.

Misu et al. elucidated the positive correlation between Selenoprotein P mRNA levels and the severity of insulin resistance in patients with T2DM.

Line 204…: multiple unnecessary repetitions in the paragraph.

(Response)

First paragraph in 2.3.1 (Zn) is necessary to explain the relationship between the severity of hepatic fibrosis and Zn status. So, the paragraph is not equivalent to unnecessary repetitions.

Line 333: pleas reformulate “A diet high fructose feeding” to a high fructose diet

(Response)

The grammatically wrong expression was corrected on the basis of a reviewer’s comment.(see the Text, red highlight).

Line 341-2: do not repeat 2 time Wilsons disease in same sentence.

(Response)

The repeat of “Wilson disease” in a sentence was avoided (see Text, red highlight).

Liggi et al. elucidated the close correlation between serum Cu level and grading of hepatic steatosis in those patients, (see the Text, red highlight)

Lines 399-421: the complete section 2.7 should be deleted as unrelated to the topic and containing not one mention about any of the 4 trace elements

(Response)

The contribution of the trace elements to dysbiosis is mentioned in “Introduction”.

So, the paragraph 2.7 “The roles of microbiota in trace elements” cannot be deleted.

Line 434: “reasonable” - please reformulate445: please delete as only very indrectl related to the topic.

(Response)

In accordance with a reviewer’s comment, “reasonable” was replaced with “plausible” (see the Text, red highlight).

Lines 437-

Reference style in the text:  question to the editors: should the authors use as thy do “.. and colleagues” or the shorter more efficient “.. et al”?

(Response)

In accordance with a reviewer’s comment, “and colleagues” was replaced with “et al.” (see the Text, red highlight).

Reviewer 2 Report

This is a very interesting review regarding trace element function in chronic liver diseases, however, I recommend the manuscript to be checked by a native speaker. There are several typos and grammatical errors, that need correction.

Include in the abstract section, which trace elements did you analyze.

The paper would benefit if at least 2 figures were included in the text. You could present, how trace elements influence the replication processes in the human body.

Introduction: Shortly describe the absorption and liver distribution processes of the trace elements described in the paper.

Describe the role of alcohol and smoking in the distribution of trace elements and its impact on hepatic fibrosis.

The reference list needs to be refreshed, there are plenty of articles from 2018, 2019 even 2020, that should be included, please cite:

Grochowski, C., Blicharska, E., Baj, J., Mierzwińska, A., Brzozowska, K., Forma, A., & Maciejewski, R. (2019). Serum iron, Magnesium, Copper, and Manganese Levels in Alcoholism: A Systematic Review. Molecules, 24(7), 1361. doi:10.3390/molecules24071361 

Author Response

Reviewer#2

This is a very interesting review regarding trace element function in chronic liver diseases, however, I recommend the manuscript to be checked by a native speaker. There are several typos and grammatical errors, that need correction.

 (Response)

Our manuscript has been already checked by a reliable native speaker before submitting. But, there were some grammatical errors in the manuscript. The errors which I found were corrected.

Include in the abstract section, which trace elements did you analyze.

(Response)

We described that our review article focused on 4 trace element (Zn, Se, Fe and Cu) in “Abstract” as shown below (see “Abstract”, red highlight)

 This review focuses on the current trends of four trace elements (zinc, selenium, iron and copper) in chronic liver diseases

The paper would benefit if at least 2 figures were included in the text. You could present, how trace elements influence the replication processes in the human body.

 (Response)

Thank you for an appropriate comment. Two figures were inserted in the text. (see Figures 1&2)

Figure 1. Relationship between zinc deficiency and inflammation, fibrosis or steatosis in the liver.

Figure 2. Putative mechanisms by which inadequate copper availability causes hepatic steatosis.

Introduction: Shortly describe the absorption and liver distribution processes of the trace elements described in the paper.

(Response)

We appreciate a proper comment. The absorption and liver distribution process was briefly described in “Introduction” as shown below (see “Introduction”, red highlight)

 Most of trace elements are absorbed from the duodenum and/or jejunum and flow out in the portal circulation by binding to the plasma proteins. These trace elements are distributed to the tissues or organs that require them.

Describe the role of alcohol and smoking in the distribution of trace elements and its impact on hepatic fibrosis.

(Response)

Thank you for reminding me of an important point. But, alcoholism was excluded from chronic liver diseases in this review article. So, we mentioned that cigarette smoking might affect the homeostasis of these trace elements in “Introduction”, as shown below. (see “Introduction”, red highlight)

A recent study revealed that even cigarette smoking might participate in the disorder of a trace element homeostasis.

The reference list needs to be refreshed, there are plenty of articles from 2018, 2019 even 2020, that should be included, please cite:

 (Response)

Several current references were cited in the text on the basis of a reviewer’s comment. (see the text, blue highlight).

  1. Diglio D.C.; Fernandes, S.A.; Stein, J., et al. Role of zinc supplementation in management of chronic liver diseases: a sysmematic review and meta-analysis. Ann Hepatol 2020, 19, 19-196
  2. Khedr, M.H.; El-Araby, H.A.; Konsowa, H.A.S, et al. Glutathione peroxidase and malondialdehyde in children with chronic hepatitis C. Clin Exp Hepatol 2019, 5, 81-87
  3. Ye, J.; Zhang, Z.; Zhu, L., et al. Polaprezinc inhibits liver fibrosis and proliferation in hepatocellular carcinoma. Mol Med Rep, 2017, 16, 5523-5528
  4. Bashandy, S.A.E.; Alaamer, A.; Moussa, S.A.A.; Omara, E.A. Role of zinc oxide nanoparticles in alleviating hepatic fibrosis and nephrotoxicity induced by thioacetamide in rats. Can J Physiol Parmacol, 2018, 96, 337-344
  5. Burk, R.F.; Hill, K.E.; Motley, A.K.; Byrne, D.W.; Norsworthy, B.K. Selenium deficiency occurs in some patients with moderate-to-severe cirrhosis and can be corrected by administration of selenite but not selenomethionine: a randomized control trial. Am J Clin Nutr 2015, 102, 1126-1133
  6. Ito, T.; Ishigami, M.; Ishizu, Y., et al. Correlation of serum zinc levels with pathological and laboratory findings in patients with nonalcoholic fatty liver disease. Eur J Gastroenteol Hepatol, 2020, 32, 748-753
  7. Swiderska, M.; Maciejczyk, M.;Zalewska, A, et al. Oxidative stress biomerkers in the serum and plasma of patients with non-alcoholic fatty liver disease (NAFLD). Can plasma AGE be a marker of NAFLD? Oxidative stress biomarkers in NAFLD patients. Free Radic Res 2019, 53, 841-850
  8. Grüngreiff, K. Branched amino acids and zinc in the nutrition of liver cirrhosis. J Clin Exp Hepatol 2018, 8, 480-483
  9.  

Round 2

Reviewer 2 Report

Please cite:

Grochowski C, Blicharska E, Baj J, et al. Serum iron, Magnesium, Copper, and Manganese Levels in Alcoholism: A Systematic Review. Molecules. 2019;24(7):1361. Published 2019 Apr 7. doi:10.3390/molecules24071361

Author Response

Reviewer#2

Comments and Suggestions for Authors

Please cite:

Grochowski C, Blicharska E, Baj J, et al. Serum iron, Magnesium, Copper, and Manganese Levels in Alcoholism: A Systematic Review. Molecules. 2019;24(7):1361. Published 2019 Apr 7. doi:10.3390/molecules24071361

(response)

In accordance with a reviewer’s comment, the reference was inserted in the text (see “Introduction” and “Reference #8, green highlight). Thus, the reference numbers after #8 were shifted (see “Text, green highlight).